# Assessment of Annual Erosion and Sediment Yield Using Empirical Methods and Validating with Field Measurements—A Case Study

**Ehsan Shahiri Tabarestani [1], Hossein Afzalimehr [1,*] and Jueyi Sui [2]**

[1] Faculty of Civil Engineering, Iran University of Science and Technology, Tehran 16846-13114, Iran; ehsan_shahiri96@civileng.iust.ac.ir

[2] School of Engineering, University of Northern British Columbia, Prince George, BC V2N 4Z9, Canada; jueyi.sui@unbc.ca

* Correspondence: hafzali@iust.ac.ir; Tel.: +0098-913-2175524

**Abstract:** To implement soil conservation approaches, it is necessary to estimate the amount of annual sediment production from a watershed. The purpose of this study was to determine the erosion intensity and sedimentation rate from a watershed by employing empirical models, including the modified Pacific Southwest Inter-Agency Committee (MPSIAC), the erosion potential method (EPM), and Fournier. Moreover, the accuracy of these empirical models was studied based on field measurements. Field measurements were conducted along two reaches of Babolroud River. Total sediment transport, including suspended load and bed load, was predicted. Bed load transport rate was measured using a Helly–Smith sampler, and suspended load discharge was calculated by a sediment rating curve. The results of this study indicate that the erosion intensity coefficient (Z) of the Babolroud watershed is 0.54, with a deposition rate of 166.469 $m^3/(km^2.year)$. Due to the existence of unusable crops, the highest amount of erosion appeared in the northern region of the watershed. The results using the EPM and MPSIAC models were compared with field measurements and indicated that both models provided good accuracy, with differences of 22.42% and 20.5% from the field results, respectively. Additionally, it could be concluded that the Fournier method is not an efficient method since it is unable to consider the erosion potential.

**Keywords:** erosion intensity; Babolroud watershed; empirical methods; field measurements; sediment rating curve

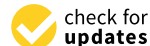



## 1. Introduction

Soil is one of the most important nonrenewable natural resources in the world. The process of soil erosion involves three distinct stages: the separation of soil particles, the transfer of particles, and the sedimentation of transported particles. Soil erosion is a global concern due to its economic and environmental impacts and has received increasingly more attention from researchers in recent years. Protecting soil erosion is crucial due to its effects on soil fertility, water quality, and flooding prediction. Due to soil erosion, millions of tons of sediments enter reservoirs and cause a decrease in their storage capacity, damage to dams, reduction of the life of dams, changes in water quality, and tremendous economic losses [1,2]. In order to implement programs to control soil erosion and reduce sedimentation, it is necessary to estimate the total volume of sedimentation and the intensity of erosion from a watershed and identify the factors influencing the erosion of the watershed. Identification of these factors will help choose appropriate approaches to control erosion and conserve natural resources [3].

There have been several methods to estimate the sediment yield from a watershed. The earliest empirical method was the universal soil loss equation (USLE) [4]. The USLE method can be used to estimate the average annual erosion rate from a watershed based on

rainfall pattern, soil characteristics, topography, and ground cover [5,6]. Some modifications have been suggested to enhance the performance of the USLE model, namely, the revised universal soil loss equation (RUSLE) [7] and modified universal soil loss equation (MUSLE) models [8]. Additional empirical methods have been named, such as Fournier, the Food and Agriculture Organization (FAO), the Pacific Southwest Inter-Agency Committee (PSIAC), the modified PSIAC (MPSIAC), and the erosion potential method (EPM). These methods have been successfully used by researchers in many watersheds [9–18].

Several researchers have estimated the amount of erosion and sediment using both the MPSIAC and EPM models for developing erosion maps in different basins in Iran; however, the Fournier method has not been evaluated in such basins. In some studies, better results have been generated using the MPSIAC model compared to those using the EPM model and vice versa, depending on the watersheds studied and their climatic and geological attributes. The level of soil erosion in a small agricultural watershed in eastern India was evaluated using the USLE model. The results showed that most of the eroded soil was deposited in rice crop check basins before reaching the outlet [12]. In one study, the total amount of soil loss and sediment yield was estimated using the RUSLE model by combining it with a geographic information system (GIS). As reported by the researchers, the sediment delivery ratio of a watershed studied in Ethiopia ranged from 0 to 0.26, and the highest value was reported for the central and eastern parts of the watershed [16].

Geographic information system (GIS) and remote sensing (RS) techniques have been successfully implemented in the assessment of erosion and sediment yield [19–21]. These techniques are cost- and time-effective and, in many cases, result in high accuracy [22–24]. Therefore, these techniques have been used all around the world as tools for the assessment and control of soil erosion and water resources.

Total sediment transport mainly consists of suspended load and bed load. Generally, the ratio of bed load to suspended load of a river is about 5–25% [25]. Due to difficulties of measuring bed load in the field, few studies have been conducted in this regard [26–28]. In one study, changes in runoff and sediment transport in the Middle Reach of the Huai River were studied using 58 years of field data [29]. Bed load sediments were calculated based on data collected using a Helley–Smith sampler in gravel bed rivers, and some of the universal bed load predictors were evaluated with the measured data. The objectives of field measurements were to evaluate the bed load transport of Babolroad River and predict suspended load with the rating curve for two sedimentation stations: Darounkola and Kerikchal Stations.

Accurate identification of eroded areas and the severity of the destruction of soil resources in a watershed will help planners optimize the management of a watershed. In this study, based on data collected in the Babolroud watershed, three empirical models, including EPM, MPSIAC, and Fournier, were evaluated. The accuracy of these models was assessed according to field measurement data at Darounkola and Kerikchal Stations.

The main difference between this study and other studies mentioned in the literature is that the results of field measurements are used for validating the empirical methods. The originality of this study is the application of field measurements for the validation of erosion intensity. To our knowledge, most of the reported studies have evaluated empirical methods by comparing the results with the amount of sedimentation reported in some stations, but they have not assessed the accuracy of field measurements.

## 2. Materials and Methods

### 2.1. Study Area

The Babolroud watershed is located between 36°0′2″ and 36°36′35″ north latitude and 52°28′40″ to 52°47′2″ east longitude in Mazandaran Province, Iran (Figure 1). Babolroud River originates from the northern front of Alborz Mountain, and it is bounded by the Talar watershed and Siahroud River in the east and Haraz River in the west. The annual flow discharge of Babolroud River is about 11 m$^3$/s, and the riverbed mainly consists of gravel and coarse particles. The drainage area of the Babolroud River watershed is about

962 km$^2$, and this watershed contains five main sub-basins. The area is mostly cold and semi-wet, with an annual precipitation depth of 782 mm and an average temperature of 14.14 centigrade. The maximum and minimum elevations of the watershed are 3677.6 and −14.8 m, respectively. A large part of the middle and southern regions is mountainous and covered by a dense forest of beech, oak, and broadleaf. In the northern region, the land is mainly used for agricultural purposes.

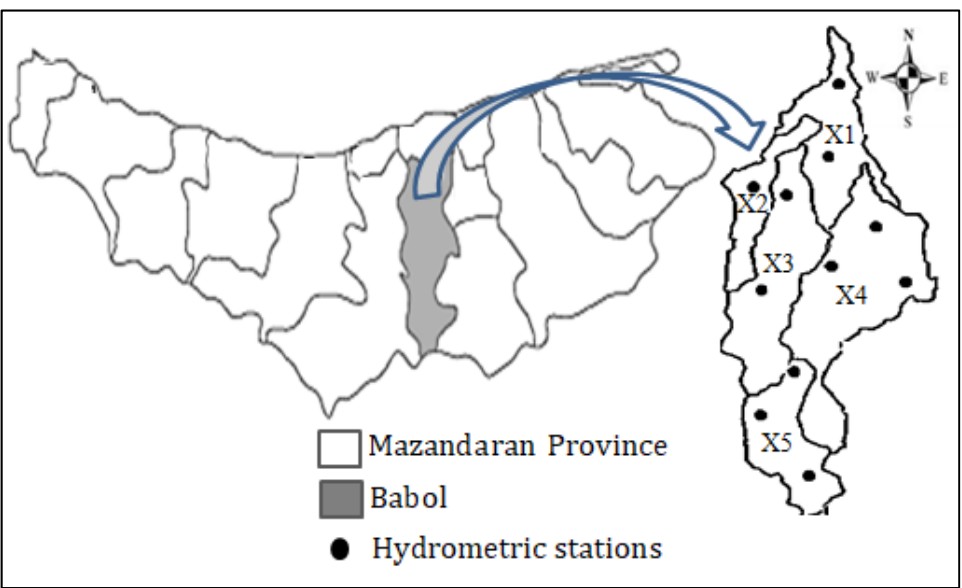

**Figure 1.** Location of the Babolroud watershed in Iran, Mazandaran Province.

### 2.2. MPSIAC Method

The PSIAC method was proposed in 1968 by the Water Management Committee of the United States to calculate the severity of soil erosion and sediment production in arid and semi-arid regions of the western United States. After modification, it was named modified PSIAC (MPSIAC) [30]. Compared to other existing empirical methods, the MPSIAC method considers more effective erosion factors and reduces the error in estimating the amount of sediment transported [1]. In the MPSIAC method, the impact of 9 effective factors with respect to soil erosion, including geology, soils, climate, runoff, topography, land cover, land use, upland erosion, and channel erosion, has been evaluated. Brief explanations of these factors are presented below.

#### 2.2.1. Surface Geology Factor (f$_1$)

The surface geology factor is related to the geologic erosion index (Y$_1$) determined by rock types and their characteristics. Loose rocks are usually easily exposed to erosion and play a key role in sediment yield. Depending on the resistance degree of rocks against erosion, the values of this factor may vary from 0 and 10 [1], which are given in Table 1.

**Table 1.** Surface geology factor scoring.

| Geounit | Description | f$_1$ |
|---------|-------------|-------|
| Qm | Swamp and marsh | 2 |
| Pel | Medium- to thick-bedded limestone | 6 |
| Mm,s,l | Marl, calcareous sandstone, sandy limestone, and minor conglomerate | 5 |
| TRJs | Dark-gray shale and sandstone | 9 |
| K2l2 | Thick-bedded to massive limestone | 5 |
| Plc | Polymictic conglomerate and sandstone | 5 |
| TRe | Bedded dolomite and dolomitic limestone | 3 |

**Table 1.** *Cont.*

| Geounit | Description | $f_1$ |
|---|---|---|
| Ktzl | Thick-bedded to massive, white to pinkish orbitolina-bearing limestone | 6 |
| Jl | Light-gray, thin-bedded to massive limestone | 5 |
| Kbvt | Basaltic volcanic tuff | 5 |
| Qft2 | Low-level piedmont fan and valley terrace deposits | 5 |

### 2.2.2. Soil Factor ($f_2$)

This factor is estimated using $16.67 \times k$, in which k is the soil erodibility factor depending on soil texture and the amount of silt, lime, gravel, and organic matter in soil [30]. The range of changes for this factor is based on soil texture, stability of aggregates, amount of lime, organic matter, ability to spread clay particles, and soil moisture. Table 2 shows the scores allocated to the types of soils in the field.

**Table 2.** Soil factor scoring.

| Type of Soil | $f_2$ | k |
|---|---|---|
| Mollisols | 6 | 0.36 |
| Rock Outcrops/Entisols | 3 | 0.18 |
| Alfisols | 7.1 | 0.43 |
| Inceptisols | 8 | 0.48 |
| Mollisols | 6 | 0.36 |
| Inceptisols | 8 | 0.48 |
| Alfisols | 7.1 | 0.43 |

### 2.2.3. Climate Factor ($f_3$)

The amount of runoff from a watershed depends on the amount and intensity of precipitation. The amount and intensity of precipitation influence the amount of erosion. This factor depends on the frequency of precipitation, the intensity of precipitation, and the period of precipitation, snow, ice, and melting. This factor is determined by $0.2 \times P_2$, in which $P_2$ is the precipitation amount during a period of 6 h with a return period of 2 years (mm).

### 2.2.4. Runoff Factor ($f_4$)

To assess the effect of runoff on soil erosion, it is necessary to consider the hydrological characteristics of the watershed, such as the specific flow of floods ($m^3 s^{-1} km^{-2}$), the specific flow with different return periods, and the hydrological groups of soils. The runoff factor is estimated by $f_4 = 0.006R + 10Q_P$, in which R is the total average runoff depth (mm) that is interpolated from measurements at the meteorological stations, and $Q_P$ is the peak special discharge ($m^3 s^{-1} km^{-2}$) determined from the peak discharge at the hydrological units.

### 2.2.5. Topography ($f_5$)

The topography factor is usually determined in accordance with the average slope of a watershed. Erosion usually increases with the slope of a watershed because of the increase in the speed of the runoff generated from a watershed. This factor can be calculated by $0.33 \times S$, in which S is the average slope of a watershed in percentage. The map of the average slope can be generated from the digital elevation model. The topography factor is very important in determining soil erosion from a watershed in the MPSIAC method by considering the score of this factor ranging from 0 to 20 [1].

### 2.2.6. Ground Cover ($f_6$)

Vegetation, litter, and rocks are types of ground cover. The presence of any of these three covers can have positive effects on preventing the watershed from soil erosion and sediment yield. The ground cover factor can be determined by $0.2 \times P_b$, in which $P_b$ is the

percentage of the bare cover accounted for in a watershed. The value of this factor ranges from −10 to 10 [1].

### 2.2.7. Land Use (f$_7$)

To determine this factor, two criteria are usually considered: the first is agricultural activities, and the second is livestock grazing status. If agricultural activities are not common at the basin level, or the watershed area is covered with dense vegetation and is less likely to be domesticated, the role of this factor in soil erosion and the sediment yield from this watershed are negative. This factor can be determined by 20−0.2 P$_C$, in which P$_C$ is the coverage of the plant canopy in percentage. The value of the land use factor ranges from −10 to 10 [1].

### 2.2.8. Upland Erosion (f$_8$)

Surface erosion in a watershed is assessed using this factor. This factor is considerably important for determining the sediment yield from a watershed, with a score ranging from 0 to 25 [1]. To assess the surface factor of soil (S.S.F.), seven aspects are considered, including soil mass movement, petiole cover, rock surface cover, pedestalling rock fragments, surface grooves, waterway form, and development of ditch erosion. This factor is estimated by 0.25 × S.S.F., in which S.S.F. is the sum of scores in the BLM method [30].

### 2.2.9. Channel Erosion (f$_9$)

Regarding the erosion from channels in a watershed, both erosion from channel banks and sediment transport by the flow are examined. Channel erosion is the result of the destruction of channel banks, which occurs mostly during floods and watery seasons. Some factors that have major effects on the deformation of the channel bed and sediment transport are the average slope of riverbeds, type of rocks along rivers and potential energy of floods. This factor ranges from 0 to 25 [1] and can be calculated by 1.67 × SSF.g, in which SSF.g is the gully erosion in the BLM method [30].

### 2.2.10. Sediment Flux

Based on the degree of impact of each factor, scores are assigned to each factor Finally, the total score is calculated, and the annual rate of sediment yield (Q$_S$) is estimated by the following equation.

$$Q_s = 38.77e^{0.0353R} \tag{1}$$

where R is the total sum of factors, and Q$_S$ is the annual rate of sediment yield from each sub-basin in m$^3$/km$^2$. In this method, the amount of soil erosion of each unit is called sediment load, which is the sum of suspended load and bed load. According to the amount of sediment produced from a watershed, the sedimentation class of each sub-basin can be obtained from Table 3.

**Table 3.** Classification of soil erosion in MPSIAC model.

| Sediment Production m$^3$/(km$^2$.year) | Erosion Intensity | Erosion Classification |
|---|---|---|
| >1429 | Very high | V |
| 476–1429 | High | IV |
| 238–476 | Moderate | III |
| 95–238 | Low | II |
| <95 | Very low | I |

### 2.3. EPM Method

The erosion potential method (EPM) was initially developed based on data collected in Yugoslavia [31]. In this method, four criteria, including watershed erosion coefficient (∅ϕ), land use coefficient (X$_a$), coefficient of rock and soil resistance to erosion (Y), and

average slope of the watershed (I), are examined. The erosion intensity coefficient (Z-factor) from a sub-basin can be determined using Equation (2):

$$Z = X_a.Y.\left(\phi + I^{0.5}\right) \tag{2}$$

The classification of sub-basins is shown in Table 4 according to the severity of erosion. In this method, the annual rate of sedimentation, $q_s$ in $\frac{m^3}{km^2.year}$, can be calculated by Equation (3):

$$q_s = T.H.\pi.Z^{1.5} \tag{3}$$

where Z is the sediment yield from a sub-basin and can be calculated from Equation (2), H is the mean annual precipitation depth (mm), and T is the temperature coefficient determined by the following equation:

$$T = \left(\frac{t}{10} + 0.1\right)^{0.5} \tag{4}$$

where t is the mean annual air temperature in the watershed (°C)

**Table 4.** Classification of erosion intensity in EPM method.

| Ranges | Erosion Intensity | Erosion Classification |
|---|---|---|
| Z > 1 | Very high | V |
| 0.71 < Z < 1 | High | IV |
| 0.41 < Z < 0.71 | Moderate | III |
| 0.2 < Z < 0.71 | Low | II |
| Z < 0.2 | Very low | I |

### 2.4. Fournier Method

The Fournier method is a rather simple method for the assessment of erosion intensity because it does not require complex calculations and experimental research. It was initially developed for estimating erosion resulting from rainfall [32]. Fournier proposed two different methods for estimating the annual rate of sediment yield from a watershed.

The first method proposed by Fournier for estimating sediment yield from a watershed is as follows:

$$\text{Log } Q_{S1} = 2.65 \text{ Log}\frac{P_w^2}{P_a} + 0.46 \text{ Log H } (\tan S) - 1.56 \tag{5}$$

where $Q_S$ is the annual sediment yield in $\frac{ton}{km^2.year}$, $P_W$ is the average precipitation depth during the rainiest month of each year in the statistical period (mm), $P_a$ is the annual precipitation depth (mm), H is the average height of the watershed (m), and S is the average slope of the watershed (degree).

The second method proposed by Fournier for estimating sediment yield from a watershed is as follows:

$$\text{Log } Q_{S2} = 2.65 \text{ Log}\frac{P_w}{P_a} + 0.46 \text{ Log}\frac{H^2}{A} - 1.56 \tag{6}$$

where A is the drainage area of the watershed (km$^2$), and other terms are similar to Equation (5). One of the main disadvantages of the Fournier methods is that they do not examine the erosion potential of the basin [32]. Therefore, if two regions are similar in terms related to Equations (5) and (6), but different in terms of geological, soil, and vegetation conditions, the estimated sedimentation using Equations (5) and (6) will be the same.

### 2.5. Research Data

Estimating the amount of sediment yield from a watershed requires various information such as topography, geology, soil, land use, rainfall, land slope, and temperature of the

area. Figure 2a shows the digital elevation map of the Babolroud watershed. The southern part of the watershed is mainly occupied by mountains, but there are plain areas with low elevations in the northern part of the watershed. Figure 2b shows the geological map of the study area. There are a lot of alluvial formations in the northern region, and there is a layer of hard and shallow rocks in the southern part. A large part of the watershed has Alfisol soil, as shown in Figure 2c. This type of soil is often observed in humid and semi-humid areas with the presence of forest. Figure 2d shows the land use map of the Babolroud watershed. One can see from this figure that the watershed is covered by dense forest and mountainous areas, orchards, forest–agricultural areas, and groves. Areas with dense vegetation, which covers a major part of the watershed, play a key role in preventing this watershed from the soil erosion process.

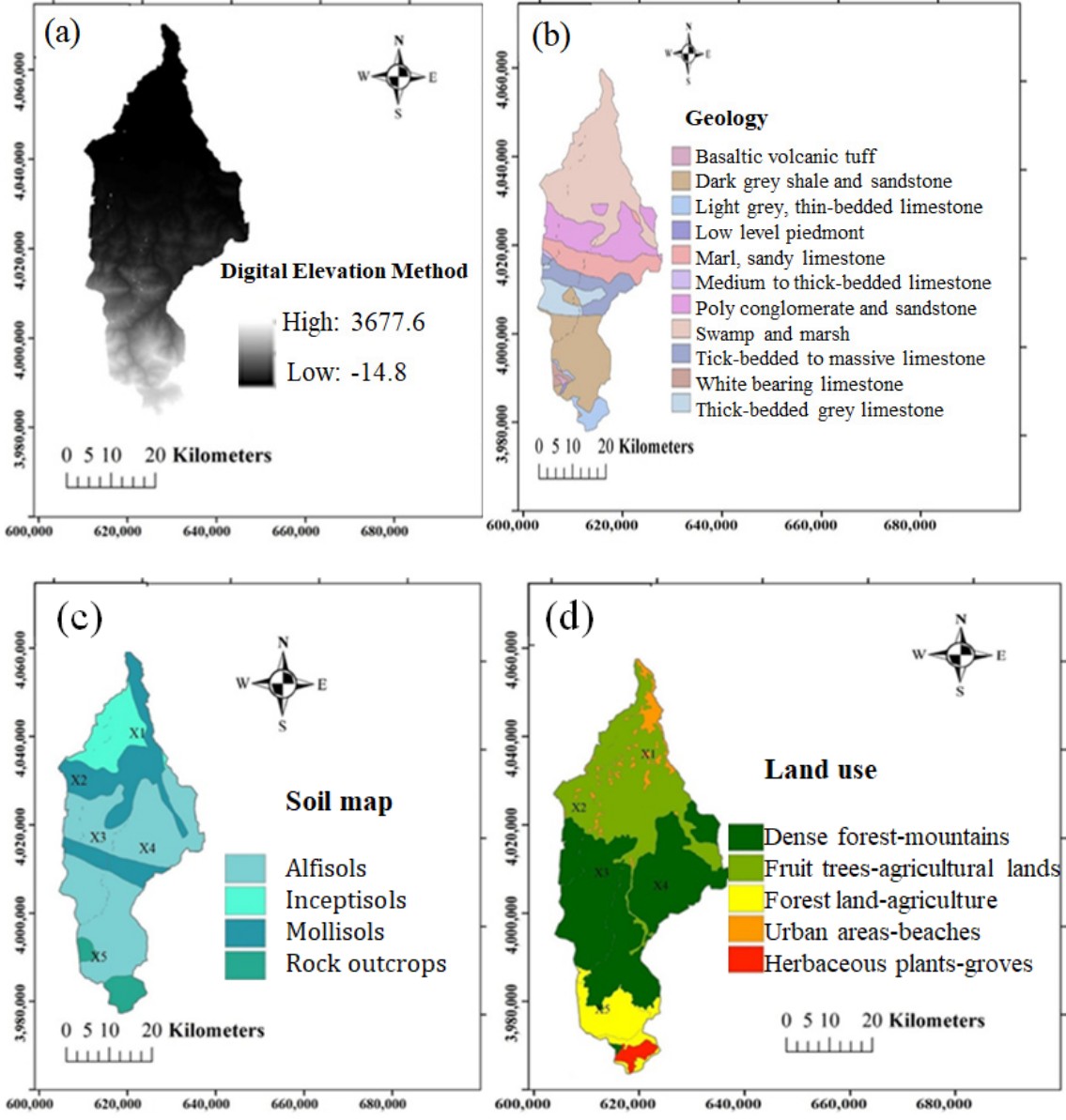

**Figure 2.** *Cont.*

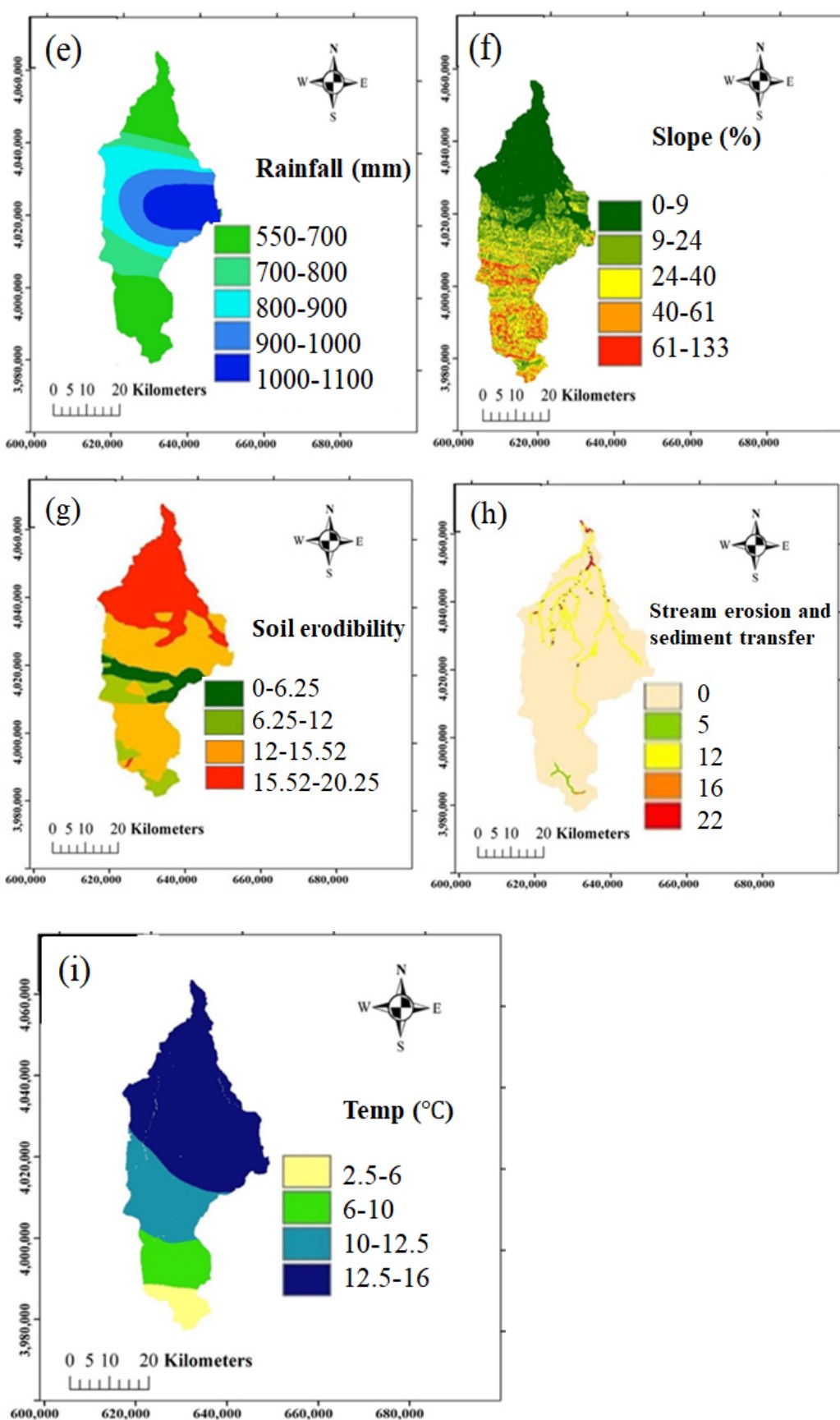

**Figure 2.** Maps of the study watershed: (**a**) digital elevation model; (**b**) geology; (**c**) soil map; (**d**) land use; (**e**) annual precipitation (mm) (**f**) slope; (**g**) upland erosion; (**h**) channel erosion; (**i**) temperature.

Based on data collected at 11 meteorological stations during the period from 2008 to 2018 in this watershed, statistical analysis was conducted. To develop a spatial map of rainfall intensity, the modified Fournier index was determined based on the meteorological information of 11 hydrometric stations from 2008 to 2018, as shown in Equation (7).

$$\text{MFI} = \sum_{i=1}^{i=12} \frac{P_i^2}{P} \tag{7}$$

MFI is the modified Fournier index, Pi is the average monthly rainfall depth, and P is the average annual rainfall depth at the "i" meteorological station. This index shows the sum of the weighted monthly rainfall depth at the "i" station. By means of this technique, the daily rainfall data are converted to monthly rainfall for the purpose of developing a raster rainfall map. The average annual precipitation depth in this watershed ranges from 500 to 1100 mm, as shown in Figure 2e. The maximum precipitation depth occurs in the east part of the watershed and gradually decreases towards the north and south of the watershed. As shown in Figure 2f, the highest slope of the watershed appears in the southern part of the basin due to its mountainous landscape, and the lowest slope in the northern part of the basin where a lot of flat plains appear. Additionally, based on temperature data collected during the period from 2008 to 2018, the temperature map of this watershed was developed. As shown in Figure 2i, in the northern part of the watershed, higher temperatures are observed, as the elevation is lower, and in the southern part of the watershed, lower temperatures are observed, as the elevation is higher. The sources and types of each influential factor are presented in Table 5.

**Table 5.** Sources and types of applied data.

| Dataset | Source | Data Type | Scale of Source Data | Derived Factors |
|---|---|---|---|---|
| Digital elevation model (DEM) | United States Geological Survey (USGS) site | Raster | 1:25,000 | Elevation, slope |
| Rainfall | 10-year meteorological data (2009–2019), Iran | Vector | 1:25,000 | Rainfall map |
| Geological map | Mazandaran Regional Water Authority, Iran | Vector | 1:100,000 | Geology, soil type |
| Land cover | Mazandaran Regional Water Authority, Iran | Vector | 1:100,000 | Land use |

### 2.6. Field Data

Field data were collected at 7 cross-sections along two straight and relatively stable river reaches of Babolroud River, named the Kelarikola and Darounkola reaches (Figure 3). The Kelarikola reach is located about 8 km downstream of the Darounkola reach. The stations for measurement sediment in the Darounkola and Kerikchal reaches are located at cross-sections D2 and K3, respectively.

All data were collected during the early spring since the maximum sediment transport occurs during the spring season with high flow. At each cross-section, the channel width (from the left bank to the right bank) was divided into equal spacing intervals of one meter. By applying Wolman's method, the median grain size of both surface and subsurface layers in the channel bed was more than 10 mm. Flow depth, flow velocity, bed slope, and bed load transport rate were determined at each cross-section. The flow velocity was measured using a current meter (BFMS-N-002-1678) with an impeller diameter of 40 mm. At each point, flow velocity was measured 3 times, and the average value of these 3 measurements was used to represent the flow velocity at this point. Along each vertical line from the channel bed to the water surface, there were about 12 points for velocity measurements. Based on measured results, the velocity profile along this vertical line was developed. In this study, a Helly–Smith sampler was used for measuring the bed load sediment. This sampler was placed on the channel bed. After a period of 60 s, particles that were larger

than the mesh size were collected in a bag. Then, all samples collected were taken to the laboratory, and grain size was obtained for each cross-section. In this study, to conduct the bed load sampling measurements, the flow cross-section was divided into 4 to 5 equal subsections. In each subsection, bed load sampling measurements were carried out and repeated 8 to 10 times at each point. The bed load samples collected at each point were analyzed. The grain size distribution of the bed load sediment for one cross-section (K3) is shown as an example in Figure 4.

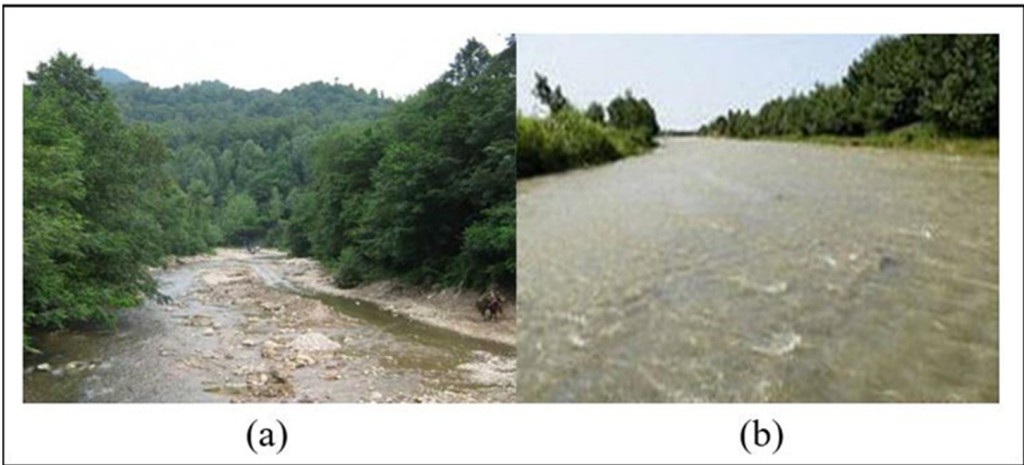

**Figure 3.** Presentation of two selected reaches: (**a**) Kelarikola reach; (**b**) Darounkola reach.

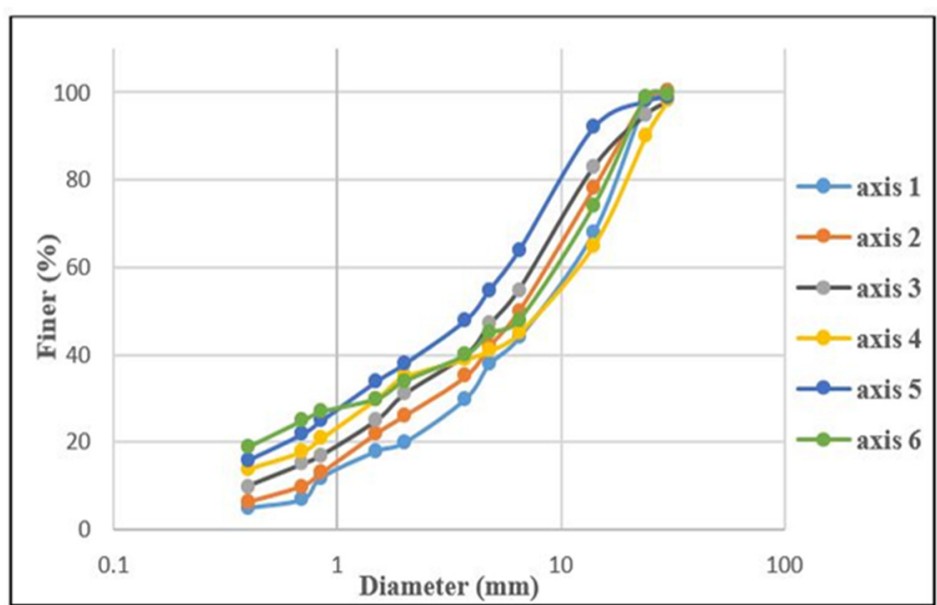

**Figure 4.** Grain size distribution of bed load at cross-section K3.

Figure 5 shows the bed load discharge at cross-sections D2 and K3. By calculating the area under the curve in Figure 5, the bed load per unit time (gr/s/m) was determined for cross-sections D2 and K3, respectively. In this figure, the bed load rate and the width of the riverbed are shown on the vertical and horizontal axes, respectively.

Bed load discharge is calculated using the following equation [33].

$$Q_b = \frac{1}{2}[(L_1 \times W_{tb1}) + L_2 \times (W_{tb1} + W_{tb2}) + \ldots + L_i \times (W_{i-1} + W_i) \qquad (8)$$

where $Q_b$ is the bed load discharge (g/s), L is the length between two points (m), and $W_{tb}$ is the dry weight per unit time and per unit width (g/s/m) and is estimated by the following equation.

$$W_{tb} = \frac{M}{W_s \times n_s \times t_s} \tag{9}$$

where M is the dry mass (gr), $W_S$ is the width of the sampler (m), $n_s$ is the number of repeated samplings, and $t_s$ is the time of the sampling duration (s).

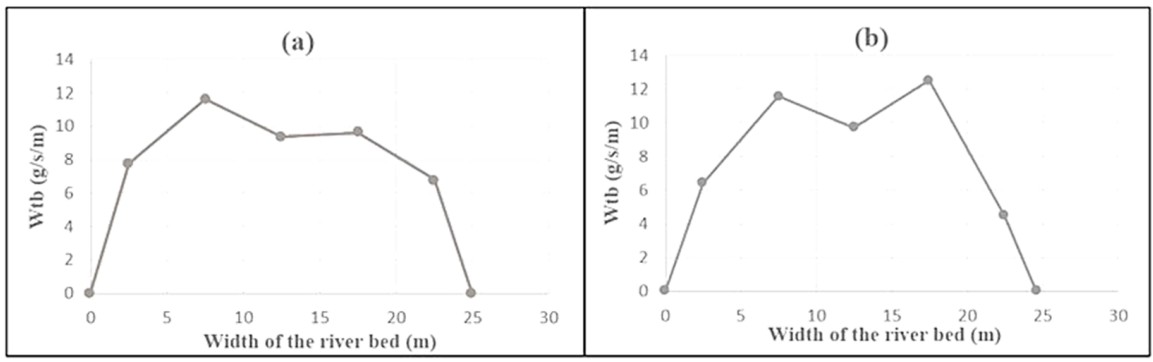

**Figure 5.** Presentation of bed load discharge at cross-section: (**a**) D2; (**b**) K3.

## 3. Results

### 3.1. Determination of Sediment Production and Erosion Class using the MPSIAC Method

As discussed above, in the MPSIAC method, there are nine effective factors impacting the rate of erosion and sediment production. By determining these factors and using Equation (1), the R parameter can be obtained, as shown in Figure 6.

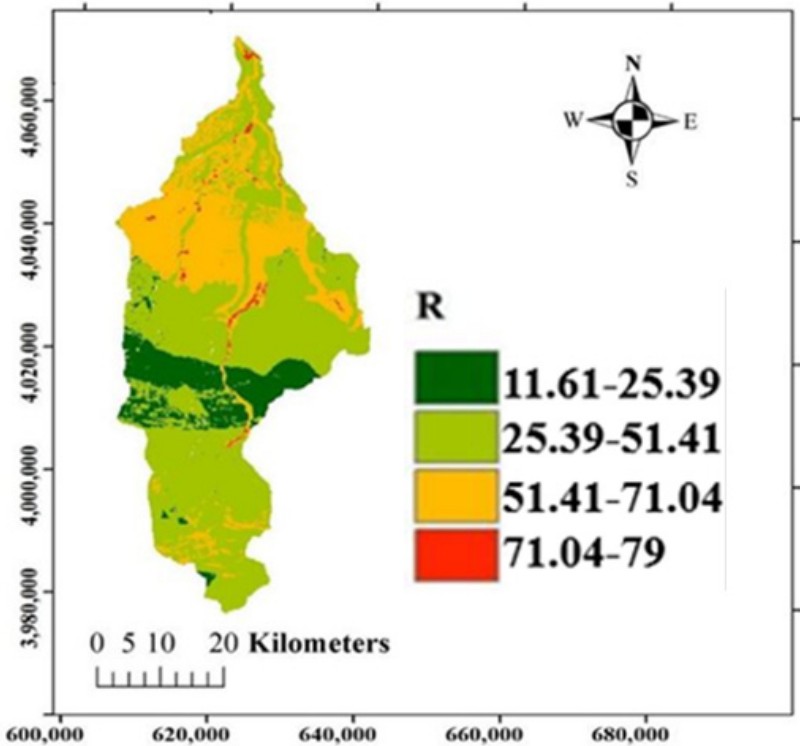

**Figure 6.** R parameter in the Babolroud watershed.

As shown in Table 6, the amount of sediment production is obtained for the sub-basins. A watershed can be classified into different erosion classes. According to Table 5, sub-basin $X_1$ located in the northern part of the watershed has a moderate erosion class, and the rest of the sub-basins are located in regions with a low erosion class. The main reason for this is the existence of uncultivated lands in the northern part of the watershed. Furthermore, the whole area with the low erosion class is attributed to the presence of dense pastures and calcareous formations and rocks with medium to high hardness in most areas. Additionally, in the middle belt of the watershed, where the soil is of Mollisol type, R values are placed in the lowest category, which indicates the importance of the type of soil in the sediment yield in this region.

**Table 6.** Amount of sediment yield and classification of sub-basins of the Babolroud basin.

| Region | R | $q_s$ $\left(\frac{m^3}{km^2*year}\right)$ | Area (km$^2$) | Qs $\left(\frac{m^3}{year}\right)$ | Class |
|--------|-----|-----------|----------|-----------|-------|
| $X_1$ | 51.9 | 238.44 | 166 | 39,582.19 | III |
| $X_2$ | 45.167 | 188.38 | 94 | 17,708.26 | II |
| $X_3$ | 36.327 | 138.25 | 226 | 31,245.38 | II |
| $X_4$ | 35.562 | 134.60 | 147 | 19,786.38 | II |
| $X_5$ | 40.053 | 157.51 | 329 | 51,821.45 | II |
| basin | 41.27 | 166.469 | 962 | 160,143.17 | II |

### 3.2. Calculation of Erosion Intensity Coefficient and Annual Sediment Yield by the EPM Method

In order to estimate parameter Z, the following information will be used, including the amount of gully and groove erosion, the type of land use, the type of formation and soil, and the slope of the area. For this purpose, the values for this required information for the Babolroud watershed were determined and are summarized in Tables 7–9. Using the values in Tables 7–9, the sub-basins of the Babolroud were classified using the EPM method based on the intensity of erosion and the amount of sediment produced, as shown in Table 10.

**Table 7.** Estimation of parameter, $X_a$ for the Babolroud watershed.

| Herbaceous Plants–Groves | Urban Areas–Beaches | Forest Land– Agriculture | Fruit Trees– Agricultural Lands | Dense Forest– Mountainous Lands |
|--------|--------|--------|--------|--------|
| 0.4 | 1 | 0.3 | 0.7 | 0.2 |

**Table 8.** Estimation of parameter, Y for the Babolroud watershed.

| Geo-unit | Description | Y |
|--------|-------------|---|
| Qm | Swamp and marsh | 2 |
| Pel | Medium- to thick-bedded limestone | 1 |
| Mm,s,l | Marl, calcareous sandstone, sandy limestone, and minor conglomerate | 1 |
| TRJs | Dark-gray shale and sandstone | 1 |
| K2l2 | Thick-bedded to massive limestone | 1 |
| Plc | Polymictic conglomerate and sandstone | 1.2 |
| TRe | bedded dolomite and dolomitic limestone | 1 |
| Ktzl | Thick-bedded to massive, white to pinkish orbitolina-bearing limestone | 1 |
| Jl | Light-gray, thin-bedded to massive limestone | 1 |
| Kbvt | Basaltic volcanic tuff | 1 |
| Qft2 | Low-level piedmont fan and valley terrace deposits | 2 |

**Table 9.** Estimation of parameter, φ for the Babolroud watershed.

| Urban Areas | Floodplain | Lowlands | Alluvial Plain | Hillside | Plateau | Crop Coverage | Forest Cover |
|---|---|---|---|---|---|---|---|
| 0.3 | 1 | 0.6 | 0.8 | 0.5 | 0.2 | 0.15 | 0.1 |

**Table 10.** Classification of sub-basins using EPM method.

| Region | Z | $R_u$ | $q_s$ $\left(\frac{m^3}{km^2.year}\right)$ | Area $(km^2)$ | $Q_S$ $\left(\frac{m^3}{year}\right)$ | Class |
|---|---|---|---|---|---|---|
| $X_1$ | 1.2 | 0.31 | 1057.36 | 166 | 175,521.76 | V |
| $X_2$ | 0.81 | 0.32 | 738.13 | 94 | 69,384.22 | IV |
| $X_3$ | 0.45 | 0.57 | 688.35 | 226 | 15,5567.1 | III |
| $X_4$ | 0.32 | 0.74 | 578.87 | 147 | 85,093.89 | II |
| $X_5$ | 0.23 | 1.25 | 236.06 | 329 | 77,663.74 | II |
| Basin | 0.54 | 0.79 | 585.47 | 962 | 563,230.71 | III |

As indicated in Figure 7a, the erosion intensity coefficients in the $X_1$ and $X_2$ sub-basins are very high and high, respectively, due to the presence of plains, orchards, and alluvial soils in the northern area of the watershed. In addition, according to the EPM method, the entire watershed is in the category of the moderate erosion intensity, which indicates that the calculated amount of erosion and sediment yield using this model are more than those using the MPSIAC model.

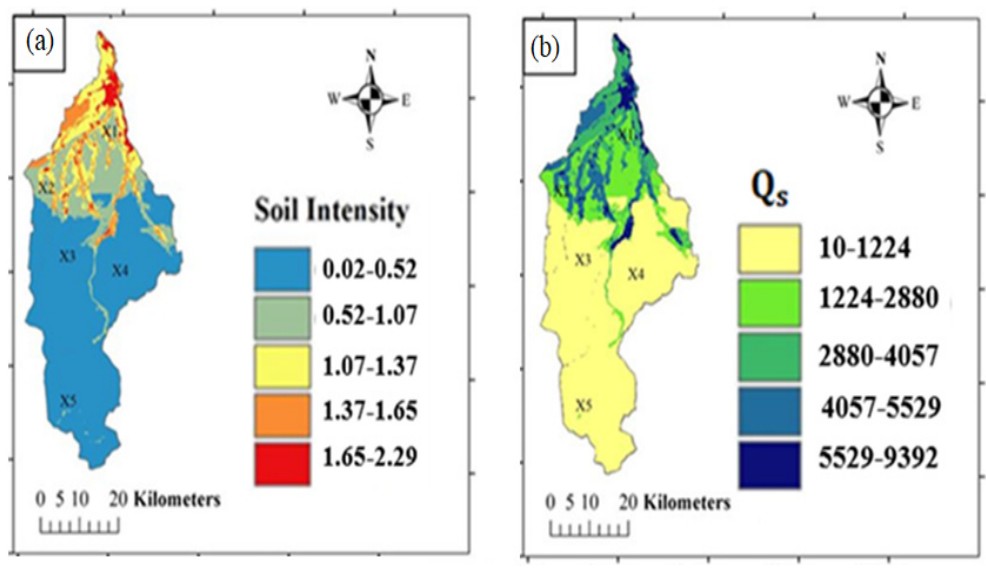

**Figure 7.** (**a**) Erosion intensity coefficient; (**b**) annual rate of sediment yield in the Babolroud watershed.

*3.3. Calculation of Erosion Intensity Coefficient and Annual Rate of Sediment Yield Using the Fournier Method*

Table 11 shows the values of the annual rate of sediment yield calculated using the first and second Fournier methods. Contrary to results using the methods discussed above, the lowest erosion rates calculated appeared in sub-basins $X_1$ and $X_2$ and the highest erosion rate in sub-basin $X_5$. Additionally, the values obtained using the first method have a high error, while the values determined using the second method are closer to the results using the other two MPSIAC and EPM methods. The main reason for the obvious difference between the results using the Fournier method and those using the EPM and MPSIAC methods is the lack of erosion potential in the study region. Figure 8 shows the annual sediment yield from sub-basins based on the first and second Fournier methods.

**Table 11.** Annual sediment yield of the Babolroud watershed based on (**a**) the first Fournier method and (**b**) the second Fournier method.

| Region | Area (km²) | $Q_{S_1}\left(\frac{Ton}{year}\right)$ | $Q_{S_2}\left(\frac{Ton}{year}\right)$ |
|--------|-----------|--------|--------|
| $X_1$ | 166 | $1.3 \times 10^9$ | 23.24 |
| $X_2$ | 94 | $4.2 \times 10^9$ | 88.36 |
| $X_3$ | 226 | $3.5 \times 10^{10}$ | 537.88 |
| $X_4$ | 147 | $3.3 \times 10^{10}$ | 327.81 |
| $X_5$ | 329 | $8.6 \times 10^{10}$ | 3911.81 |
| Basin | 962 | $6.7 \times 10^{11}$ | 4889.1 |

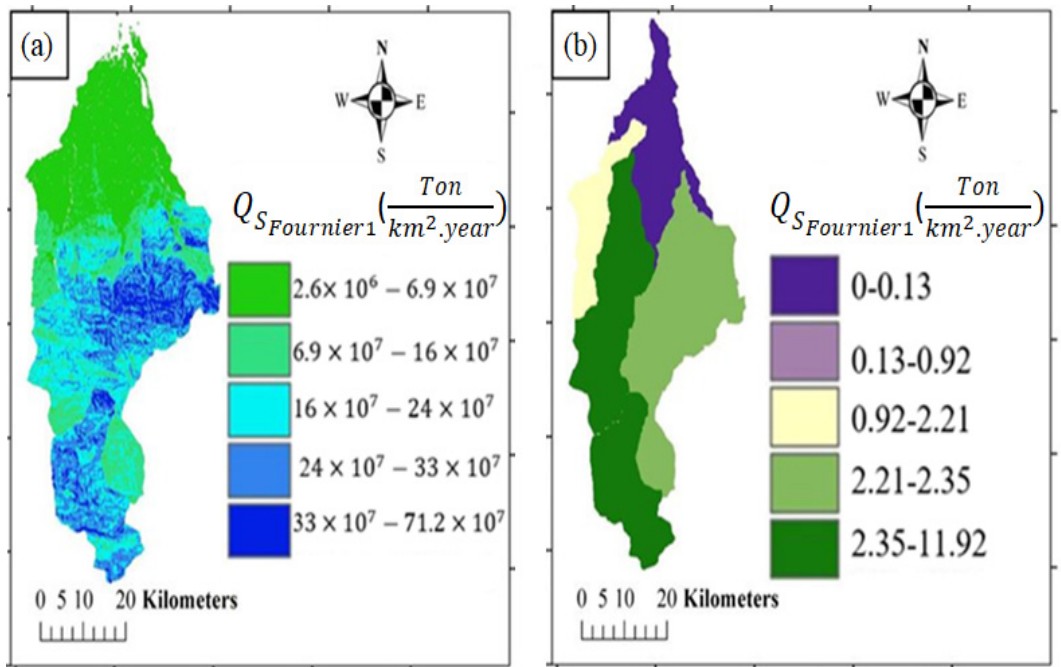

**Figure 8.** Annual sediment yield based on (**a**) the first Fournier method and (**b**) the second Fournier method.

*3.4. Verification of Model with Field Measurements*

To verify the results obtained from the models, field data collected at the two sedimentation stations, Darounkola (at D2 cross-section) and Kerikchal (at K3 cross-section), are used. Table 12 summarizes some parameters collected at these two stations in Babolroud River.

**Table 12.** Measured parameters in Babolroud River.

| Cross-Section | Slope S (m/m) | Width W (m) | Hydraulic Depth h (m) | Mean Flow Velocity $U_{eq}$ (m/s) | Bed Load Transport Rate $q_b$ (ton/day) | Discharge q (m²/s) |
|---------------|---------------|-------------|------------------------|------------------------------------|------------------------------------------|---------------------|
| D1 | 0.0071 | 23.3 | 0.395 | 0.989 | 0.634 | 0.391 |
| D2 | 0.0077 | 25 | 0.391 | 1.094 | 0.717 | 0.428 |
| D3 | 0.0056 | 24.7 | 0.432 | 0.965 | 0.702 | 0.417 |
| K1 | 0.0009 | 28 | 0.385 | 1.093 | 0.580 | 0.421 |
| K2 | 0.0007 | 25.2 | 0.521 | 0.95 | 0.762 | 0.496 |
| K3 | 0.0058 | 24.6 | 0.570 | 0.926 | 0.736 | 0.528 |
| K4 | 0.0078 | 25.4 | 0.561 | 0.862 | 0.612 | 0.484 |

On the other hand, the hydrometric data set, including the suspended sediment concentration (C), water discharge (Q), and average rate of suspended load (milligrams/liter), at the four gauging stations in Babolroud River were collected by the Mazandaran Regional

Water Authority. As reported by other researchers, based on long-term field measurement data, the relationship between discharge and sediment transport can be employed to determine the features of sediment transport and assess the changes in runoff and sediment yield from watersheds [34–36].

Suspended sediment concentration samples at these stations in this river were collected several times a year, including a data set for the period from 2008 to 2018. With this data set, the sediment rating curve can be plotted (Figure 9). Similar to the results of other researchers [34–36], a regression analysis was performed to obtain a power function between suspended sediment discharge ($Q_S$) and water discharge (Q) as follows:

$$Q_s = 23.539Q^{1.1436} \tag{10}$$

$$R^2 = 0.678 \tag{11}$$

where $Q_S$ is the suspended sediment discharge (ton/day), and Q is water discharge (m$^3$/s).

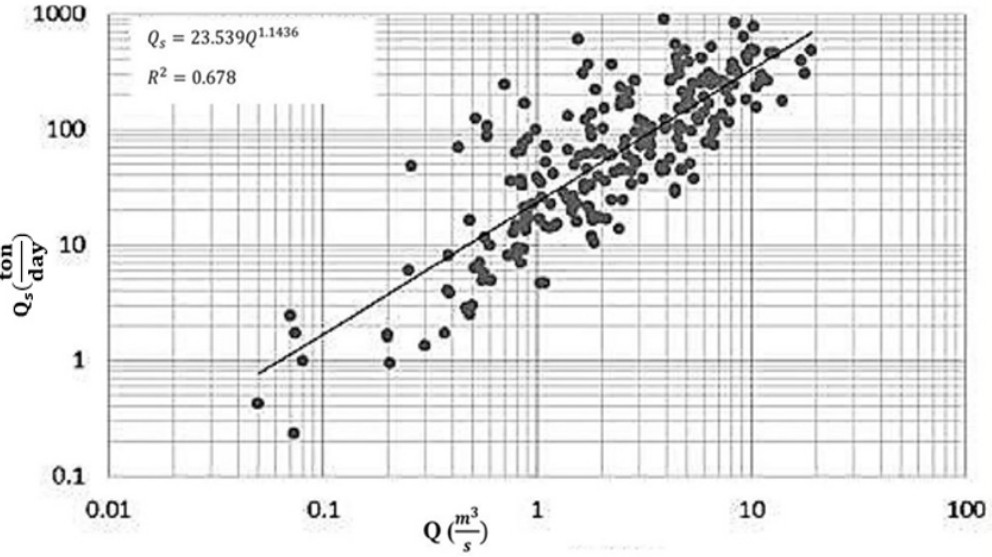

**Figure 9.** Sediment rating curve for Babolroud River.

Applying the sediment rating curve for Babolroud River, the discharge values of suspended sediment at both Darounkola and Kerikchal Stations were calculated. Table 13 shows results of flow discharge, suspended sediment discharge ($Q_S$), and bed load sediment discharge ($Q_b$) at all cross-sections along these two river reaches.

**Table 13.** Flow and sediment discharge at all cross-sections.

| Cross-Section | Q (m$^3$/s) | $Q_s$ ($\frac{ton}{day}$) Calculated | $Q_b$ ($\frac{ton}{day}$) Measured |
|---|---|---|---|
| D1 | 9.1 | 294.136 | 14.772 |
| D2 | 10.7 | 353.990 | 17.925 |
| D3 | 10.3 | 338.897 | 17.339 |
| K1 | 11.8 | 395.906 | 16.240 |
| K2 | 12.5 | 422.877 | 19.202 |
| K3 | 13 | 442.276 | 18.106 |
| K4 | 12.3 | 415.148 | 15.545 |

Table 14 shows a comparison of the results calculated using the EPM and MPSIAC methods compared to those of field measurements at both Darounkola and Kerikchal Stations. Due to the large differences between the results using the Fournier method and

those using the other two methods (EPM and MPSIAC methods), the results of the Fournier model were refused in this study.

**Table 14.** Results predicted by MPSIAC and EPM methods compared to the field measurements.

| | | Field Measurements | | | MPSIAC | EPM |
|---|---|---|---|---|---|---|
| Station | $\mathbf{Q}$ $\left(\frac{\mathbf{m}^3}{\mathbf{s}}\right)$ | $\mathbf{Qs}$ Suspended $\left(\frac{\mathbf{ton}}{\mathbf{day}}\right)$ | $\mathbf{Qs}$ Bed $\left(\frac{\mathbf{ton}}{\mathbf{day}}\right)$ | $\mathbf{Qs}$ Total $\left(\frac{\mathbf{ton}}{\mathbf{day}}\right)$ | $\mathbf{Qs}$ Total $\left(\frac{\mathbf{ton}}{\mathbf{day}}\right)$ | $\mathbf{Qs}$ Total $\left(\frac{\mathbf{ton}}{\mathbf{day}}\right)$ |
| Darounkola | 10.7 | 353.99 | 9.86 | 371.915 | 287.38 | 248.272 |
| Kerikchal | 13 | 442.276 | 18.106 | 460.382 | 376.24 | 520.72 |

As shown in Figure 10, at Darounkola Station, results calculated using the MPSIAC model are closer to the results of field measurements compared to results using the EPM. At Kerikchal Station, however, results calculated using the EPM are slightly better than results calculated using the MPSIAC model. Overall, the calculation error using the EPM method for these two sedimentation stations is 22.42%, and the MPSIAC model is 20.5%. Thus, compared to the results using the EPM method, the MPSIAC method can be used to predict sediment yield closer to results from rating curves generated based on field measurements at these two stations, indicating the better performance of the MPSIAC model in the Babolroud watershed.

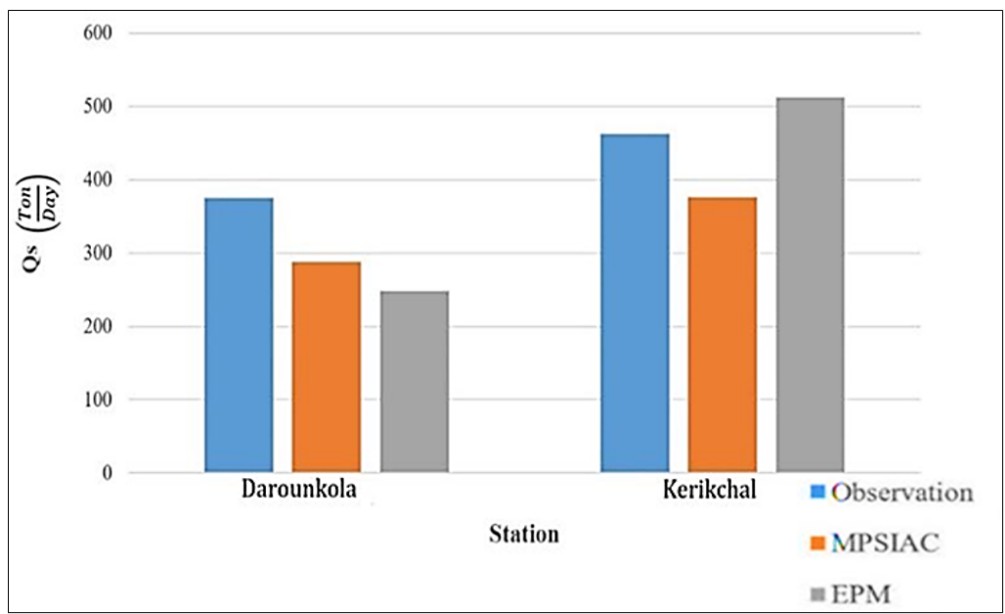

**Figure 10.** Comparison between the results predicted by both MPSIAC and EPM models to those of field measurement at Darounkola and Kerikchal Stations.

## 4. Discussion

Empirical methods for assessing soil erosion can be worked out with available inputs to estimate the sediment yield for areas exposed to high erosion risk. This research assessed three empirical methods for soil erosion estimation, including MPSIAC, EPM, and Fournier methods integrated with a GIS 10.5 model, compared to the results of field measurements in the Babolroud watershed, Iran.

The results of this research show that the sediment yield estimated by the MPSIAC method is closer to the results of field measurements at the two sedimentation stations, compared to the results by using the EPM method. This finding indicates that the MPSIAC model should be preferred to apply in the Babolroud watershed. As also reported by other researchers, the MPSIAC method has better performance in comparison with the EPM

method in the Talar watershed, which is also located in Mazandaran Province and has similar geographic characteristics to the watershed studied [15]. Nevertheless, the results of a study in the DEZ watershed showed that the EPM model generates better results than those using the MPSIAC model [10]. It should be noted that this watershed is located in the south of Iran and has different climatic and geological features in comparison to those located in Mazandaran Province. In another study in Khorasan Province in Iran, it was proved that the MPSIAC method underestimates the sediment yield [13], which is in line with the findings of the current research. Moreover, it can be concluded that the Fournier model should not be considered for areas with the same characteristics as those of the Babolroud watershed, as shown in a previous study, where this method had no efficiency in predicting erosion intensity [37]. The results of another study that was conducted in the mountainous region with a semi-arid climate suggested that the MPSIAC model is suitable for predicting the annual average sediment yield of Iranian watersheds under similar conditions [38]. In another area in the east of Iran that has a semi-arid climate, the application of the PSIAC and MPSIAC models was evaluated. The results showed that the calculated annual sediment yields in most parts using both models agree well with those of field observations [39].

The application of field measurements in determining the bed load of fluvial rivers was proved in several studies [40–43]. In this regard, bed load sediments have been estimated with reasonable accuracy. The sediment rating curve is also an accurate method for calculating suspended load in rivers [44–46]. Resultantly, field measurement can be a precise method for the validation of employed numerical methods.

It is suggested to apply these empirical models in further research for determining the annual sediment yields from other watersheds with different geographic features and compare the results with those of the present study.

## 5. Conclusions

Accurate estimation of the amount of erosion and sedimentation is impossible in all areas due to technical, protective, and economic reasons. Therefore, the most appropriate method is to estimate the amount of erosion and sediment yield, which requires knowledge of the erosion mechanisms and the factors affecting them. On the other hand, choosing the appropriate model for each region requires the evaluation of the accuracy of different methods by comparing their results with the measured values in a watershed. In the present study, the EPM, MPSIAC, and Fournier methods were used to estimate the amount of erosion and sediment yield. Field measurements were also carried out at seven cross-sections along two reaches of Babolroad River, and the bed load transport rate was calculated. Suspended sediment discharge was calculated by applying the sediment rating curves. The total sediment load was determined based on field measurement data. The results of empirical methods showed that the erosion status of the area is in the moderate erosion class, and it is necessary to carry out watershed management and soil protection in this area. The highest erosion intensity is in the $X_1$ sub-basin, mainly due to the lack of uncultivated land in the northern part of the watershed. The southern areas were less exposed to erosion due to the layer covered by hard and shallow rocks, forest, and mountain coverings. A comparison of results by using both empirical methods and field measurements at the two sedimentation stations on Babolroud River showed that both the EPM and MPSIAC methods can better predict the intensity of erosion and sediment production from the Babolroud watershed compared to the Fournier method. The total sedimentation of Darounkola Station was 371.915, 287.38, and 248.272 ton/day for field measurement, MPSIAC, and EPM, respectively. Additionally, the values for Kerikchal Station were 460.382, 376.24, and 520.72 ton/day for field measurement, MPSIAC, and EPM, respectively. The calculation error for these two sedimentation stations was 22.42% and 20.5% for the EPM and MPSIAC methods, respectively, indicating the better performance of the MPSIAC model in the Babolroud watershed.

**Author Contributions:** E.S.T.; conceptualization, methodology, software, measurements, and writing, H.A.; supervision, methodology, and validation, J.S.; methodology and writing reviewing. All authors have read and agreed to the published version of the manuscript.

**Funding:** This research received no external funding.

**Institutional Review Board Statement:** Not applicable.

**Informed Consent Statement:** Not applicable.

**Data Availability Statement:** Data will be available upon request to the authors.

**Acknowledgments:** Special thanks to Mazandaran Regional Water Authority for providing data and information of this article.

**Conflicts of Interest:** The authors declare no conflict of interest.

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
