# Peer review of "Assessment of Annual Erosion and Sediment Yield Using Empirical Methods and Validating with Field Measurements—A Case Study"

_water, doi:10.3390/w14101602_

Round 1
Reviewer 1 Report
n this article the results of field measurements of erosion processes were comprised with results obtained by the empirical methods. The investigations were conducted in the basin of the Babolroud River. In the northern part of the basin, the land is mainly used for agricultural purpose. Therefore this research has a great importance for ecology and environmental protection.
As the recomendation, it will be useful to give the results of empirical calculation for other river basin from other geographical zones. It is important also to compare these results with erosion processes in the other types of river valleys from other regions on the base the same parameters and calculations.
Reviewer 2 Report
- The work selected three empirical methods, MPSIAC, EPM and Fournier to assess “annual erosion and sediment yield”. Since these three methods are not commonly used methods, it is necessary to elaborate the selection basis in the Introduction and theoretical background in the Method part. To my understanding, the authors used these methods to calculate the slope erosion rate and riverine sediment yield. If so, please introduce how each method calculates these two terms separately.
In MPSIAC method, how is slope erosion rate calculated? Do you refer to “upland erosion” in 2.2.8 as slope erosion? It is better to illustrate how to obtain the upland erosion and channel erosion in detail which is important for readers to understand this method.
In EPM method, the sediment yield was calculated using a series of coefficients, watershed erosion coefficient (∅), land use coefficient (??), coefficient of the rock and soil resistance to the erosion, and the average slope of a watershed (I). But how did you calculate these coefficients? The theoretical background is important to understand the simulation capacity of these methods.
In Fournier method, please note clearly the source of the citation.
- There are many concepts about sediment load in the text, i.e., annual rate of sediment yield, coefficient of erosion, sediment yield, specific production of sediment, erosion intensity coefficient, degree of sedimentation, many of which are not professional terms. The author needs to make clear the differences and relations between these concepts and try to use internationally accepted variable names and standard units. In addition, the variables involving erosion or sediment yield are different in the three methods, so how can you accurately compare their results?
- Did the three models calculate bed load or suspended load? It needs to be clear. The author introduced that the bed load was observed in field. So what is the data source of the suspended sediment load?
- Line 251 “Based on data collected at 11 hydrometric stations during the period from 2008 to 2018 in this watershed, statistical analysis have been conducted.” Please indicate the number of weather stations and the number of hydrological stations separately and mark their locations on the map. Please add information such as the types and time scales of hydrometric data used.
- In 2.5 research data, it is better to list the data sources and resolution used in the manuscript.
- In Figure 1, what are the five regions (X1~X5) divided? What is the purpose?
- Line 196, “Most suspended 195 sediments and bed loads are quartz with a specific gravity of 2650 kg/m3.“ Why do you say that?
- It is suggested to complete the information about the caption, unit, legend of the figures and tables to improve its readability. For example, in table3 how does the variable correspond to the variable in MPSIAC method? What is “Scores of sediment production”? In Table 4, why do you use both average value and range to determine erosion intensity? In Table 5, which variable is “degree of sedimentation”, and how is it calculated?
Author Response
Please see attached discussion letter.
